# Prenatal stress assessment using heart rate variability and salivary cortisol: A machine learning-based approach

Rui Cao[1], Amir M. Rahmani[2,3,4], Karen L. Lindsay[5,6]*

1 Department of Electrical Engineering and Computer Science, University of California, Irvine, California, United States of America, 2 Department of Computer Science, University of California, Irvine, California, United States of America, 3 School of Nursing, University of California, Irvine, California, United States of America, 4 Institute for Future Health (IFH), University of California, Irvine, California, United States of America, 5 UCI Susan Samueli Integrative Health Institute, Susan & Henry Samueli College of Health Sciences, University of California, Irvine, California, United States of America, 6 Department of Pediatrics, Division of Endocrinology, University of California, Irvine, California, United States of America

* kllindsa@hs.uci.edu

## Abstract

### Objective

To develop a machine learning algorithm utilizing heart rate variability (HRV) and salivary cortisol to detect the presence of acute stress among pregnant women that may be applied to future clinical research.

### Methods

ECG signals and salivary cortisol were analyzed from 29 pregnant women as part of a crossover study involving a standardized acute psychological stress exposure and a control non-stress condition. A filter-based features selection method was used to identify the importance of different features [heart rate (HR), time- and frequency-domain HRV parameters and salivary cortisol] for stress assessment and reduce the computational complexity. Five machine learning algorithms were implemented to assess the presence of stress with and without salivary cortisol values.

### Results

On graphical visualization, an obvious difference in heart rate (HR), HRV parameters and cortisol were evident among 17 participants between the two visits, which helped the stress assessment model to distinguish between stress and non-stress exposures with greater accuracy. Eight participants did not display a clear difference in HR and HRV parameters but displayed a large increase in cortisol following stress compared to the non-stress conditions. The remaining four participants did not demonstrate an obvious difference in any feature. Six out of nine features emerged from the feature selection method: cortisol, three time-domain HRV parameters, and two frequency-domain parameters. Cortisol was the strongest contributing feature, increasing the assessment accuracy by 10.3% on average

**Data Availability Statement:** All relevant data have been uploaded to the online data repository DRYAD (https://doi.org/10.7280/D12Q4P).

**Funding:** This research was funded by the National Institute of Health grant number K99/R00 HD-096109 (KL).

across all five classifiers. The highest assessment accuracy achieved was 92.3%, and the highest average assessment accuracy was 76.5%.

## Conclusion

Salivary cortisol contributed a significant increase in accuracy of the assessment model compared to using a range of HRV parameters alone. Our machine learning model demonstrates acceptable accuracy in detection of acute stress among pregnant women when combining salivary cortisol with HR and HRV parameters.

## Introduction

Stress is a body's physiological response to one or more stimuli that have disrupted its mental or physical equilibrium [1]. In contrast to the environments in which our stress reaction system evolved, the growing mental burden from societal expectations and daily workload in the present day contributes to pervasive, chronic stress [2]. In a comprehensive survey conducted in 2017, up to 60 percent of employees across 35,000 organizations worldwide reported high levels of stress, and 35 percent reported constant but manageable stress levels [3]. Prolonged stress is usually associated with depletion of overall health [4], resulting in a higher likelihood of disease, including psychological illnesses, heart disease, asthma, obesity, and diabetes [5]. Many studies have also shown that stress can lead to various maladaptive health behaviors such as smoking, poor sleep, and unhealthy eating habits [6].

Maternal stress in the context of pregnancy may be particularly harmful due to the transmission of stress signals to the developing fetus. Pregnant women may encounter various stressors that exceed those of everyday life for non-pregnant people including worries about the wellbeing of themselves and their developing baby, concerns about delivery, and financial or other socioeconomic stressors associated with a growing family. The Covid-19 pandemic in the last two years is another factor that significantly contributes to elevated maternal stress [7]. These stressors have been shown to be associated with preterm birth, low birth weight, risk of gestational hypertension, and adverse health and behavioral outcomes in offspring [8–10]. In order to design effective interventions that may help to mitigate the adverse effects of maternal stress on pregnancy and offspring health outcomes, there is first a need to understand the optimal approach for assessing stress in pregnancy.

Stress assessment with high accuracy is challenging as stress is related to many different factors. The most common method of assessing current or recent psychological stress is through self-report questionnaires [11]. Although questionnaires and interviews are practical and enable researchers to collect subjective information from a large number of participants, these methods suffer from multiple disadvantages including recall bias, social desirability bias, and ignoring some questionnaire items. Studies that aim to capture changes in stress by administering repeat questionnaires over a short time span can cause survey fatigue [12]. Furthermore, inter-individual variability in how questionnaire items are interpreted may reduce the validity of some questionnaires [13]. In standard clinical practice with pregnant women, stress assessment surveys are not typically used due to time constraints and the various limitations of these methods as noted above. As research on objective stress characterization using real time, non-invasive technology evolves, there is potential for these techniques to be translated to clinical settings so that clinicians and/or their pregnant patients may benefit from momentary feedback on patients' physiological state. In turn, this could facilitate implementation of simple,

effective stress management techniques that could reduce the adverse effects of stress on health outcomes for mother and baby.

Objective stress assessment methods using ubiquitous sensing and machine learning technologies are therefore preferable in human studies to more reliably capture stress reactivity or changes in stress over time. Statistical methods can measure stress according to one or two physiological features at a time. However, machine learning approaches can assess stress objectively by fusing multiple physiological features, and even multiple modalities, thereby increasing efficiency and accuracy. Psychological stress first activates the autonomic nervous system (ANS), which can be detected through fluctuations in biological signals such as stress hormones or heart rate activity [14].

Cortisol, a glucocorticoid stress hormone, is one of the primary biomarkers of psychological stress [15]. During pregnancy, cortisol plays a crucial role in supporting fetal development [16]. Maternal cortisol levels may be increased up to four times compared to non-pregnant individuals [17], which has a positive influence on fetal neural development [18]. However, excessive prenatal stress results in fetal exposure to heightened circulating cortisol levels, which has negative consequences for neurodevelopmental outcomes and may contribute to preterm delivery [19]. Although cortisol is readily detectable in saliva in the unbound form and salivary cortisol levels correlate strongly with serum levels [20], assessing stress via cortisol measurement in pregnancy has several challenges. As cortisol levels increase with advancing gestation, studies must aim to standardize the gestational timing of sample collection across subjects. Moreover, stress assessment relying on self-obtained saliva collections in free-living situations may be inaccurate as cortisol is very sensitive to changes across time of day and time of awakening [21].

Heart rate variability (HRV) is another important avenue for stress research but has been understudied in the context of pregnancy. HRV is a measure of cyclical variations in beat-to-beat intervals. It contains time-domain, frequency-domain as well as nonlinear parameters. Greater variation in beat-to-beat intervals indicates lower stress activity in the ANS. Therefore, lower levels of time-domain HRV features are indicative of higher stress [22]. Certain patterns in frequency-domain HRV parameters (e.g., low-frequency band and high-frequency band) are associated with stress exposure [23]. The ratio of the low-frequency band and the high-frequency band is also associated with self-reported maternal depression and stress in pregnancy [24]. HRV can be easily measured using non-invasive wearable technology, making it an attractive physiological measure in both lab-based and free-living research settings. However, little is known about the reliability of HRV parameters as a measure of stress and stress reactivity in pregnancy, or how HRV performs compared to cortisol levels to characterize stress.

The aim of this study is to develop an objective, multi-feature algorithm to detect the presence of acute stress among pregnant women following standardized stress exposure, using HRV parameters, salivary cortisol, and machine learning with the help of a filter-based feature selection method. We also determine the contribution of including salivary cortisol measures to increase the accuracy of the assessment model. To the best of our knowledge, this research is the first effort to leverage the association between cortisol and HRV in building objective stress assessment methods.

## Methods

### Study design

This is a secondary analysis of cortisol and HRV data collected from a cross-over study that aimed to assess the effects of superimposed psychological stress on the postprandial metabolic response to a standardized breakfast meal during pregnancy. The study was a cross-over

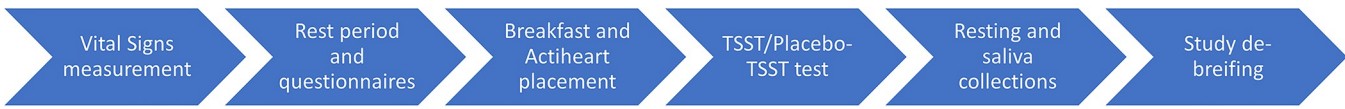

**Fig 1. Data collection protocol for each study visit.**

design involving two visits to the clinical research facility, during which participants underwent either a control non-stress task or an acute psychosocial stress challenge task, with a 1 to 2 week washout period between visits. The visit order was the same for all participants, with the control task on the first visit and stress task on the second, in order to reduce the likelihood of higher anticipatory stress when returning for visit 2 if the stress task was undertaken on visit 1. The University of California, Irvine (UCI) institutional review board approved this study, and all participants provided written informed consent.

## Participants and recruitment

Pregnant women were recruited to the study between February 2018 and March 2020. Women were eligible if they were of Hispanic ethnicity, aged 18–40 years, had a pre-pregnancy BMI 25.0–34.9 $Kg/m^2$, carrying a singleton pregnancy, less than 30 week's gestation, non-diabetic with a normal result on the standard glucose challenge test at 24–28 weeks, non-smoker, and fluent in either English or Spanish. Study exclusion criteria were non-Hispanic ethnicity, aged <18 or >40 years, pre-pregnancy BMI <25.0 or ≥35.0 $Kg/m^2$, carrying more than one baby, >30 week's gestation, diabetes (including gestational diabetes mellitus), hypertension, preeclampsia, diagnosis or treatment of any other condition that may disrupt metabolic, endocrine or immune function, diagnosis of a current psychiatric disorder or use of psychotropic medications, unwillingness to attend two study visits or consume the milkshake drink on each visit. The parent study restricted eligibility to women of Hispanic ethnicity to have a homogenous cohort of participants known to experience higher levels of socio-cultural stressors. Homogeneity among participants is important to limit inter-individual variability that could influence metabolic biomarkers that are primary outcome measures for the parent study. Recruitment took place in-person at UCI Health-affiliated obstetric clinics in Orange County, California, or by passive measures through the dissemination of brochures or recruitment emails.

## Data collection

The data collection protocol is shown in Fig 1. Participants arrived at the clinical research facility in the morning (8–9 am) following an overnight fast. Their cell phone and any other electronic devices in their possession were stored in a secure location for the duration of the visit to avoid potential distractions that may influence stress levels. Participants were asked to report their time of awakening on the morning of each visit. On visit 1, usual stress levels over the past month were assessed by the Perceived Stress Scale [25].

A baseline saliva sample was collected before consuming a milkshake drink, which was the same at both visits. Participants then stood upright while an Actiheart electrocardiograph monitor (CamNtech Ltd.) was placed on the chest to continuously measure heart rate and inter-beat interval. A second saliva sample was collected at 15 minutes after the baseline sample, followed by the task period (15-minute duration), and post-task saliva sample collection immediately after the task (30 minutes post-baseline sample). Subsequent saliva sample collections occurred at 45-, 60-, 90-, and 120-minutes post-baseline. Only water was provided and no other food or drink was consumed until the end of the study visit. Apart from the

20-minute period from the time of Actiheart placement until the end of the task, participants remained comfortably seated throughout the visit with neutral reading material.

The 15-minute control non-stress task during visit 1 was performed with a friendly research staff member already familiar with the participant. This involved a relaxed conversation on topics such as recent or upcoming holiday plans, movies, TV shows, or books they enjoyed or selecting baby names. The researcher ended the task period by announcing the time for the next saliva sample collection.

The Trier Social Stress Test (TSST) was used during visit 2 as a standardized, validated performance task designed to activate the hypothalamic-pituitary-adrenal (HPA) axis and the sympathetic nervous system to evoke a measurable physiological stress response [26]. This laboratory-based psychosocial stressor incorporates elements of unfamiliarity, uncontrollability, and a threat to self-esteem to induce a short-term psychological and physiological stress response. The TSST has been previously administered among pregnant populations without any adverse effects [27]. It consists of 5 minutes of speech preparation, 5 minutes of speech performance under the stern observation of an evaluative committee while being video-taped, and a 5-minute complex, mental arithmetic task. This task was performed in a different room to other study procedures and with two research personnel who were unknown to the participant to increase sensations of unfamiliarity.

Saliva samples were collected using Salimetrics oral swabs, which were placed under the tongue for 2 minutes at each collection time point. Saturated swabs were inserted into a Salivette tube (Sarstedt) and centrifuged at 1500g for 15 minutes to extract saliva, which was then aliquoted and stored at -80˚C. Saliva samples were assayed for cortisol at the Institute for Interdisciplinary Salivary Bioscience Research, UCI, using a commercially available enzyme immunoassay (Salimetrics, LLC). The assay uses 25 microliters of saliva per determination, has a lower limit of detection of 0.003 µg/dl, a standard curve range from 0.012 to 3.0 µg/dl, and average intra- and inter-assay coefficients of variation less than 5% and 10%, respectively. Assays were performed in duplicate and the average of the duplicate was used in statistical analyses. Outliers were assessed by detecting data points greater than 3SD from the mean on samples from both visits and outlying values were excluded. To account for inter-individual variation in cortisol values influenced by time of awakening on the mornings of the study visits, we used normalized cortisol variables, adjusted for time (minutes) interval from awakening until time of arrival at the visit, as the input for statistical analysis and the assessment model.

## Data processing and analysis

**Actiheart data.** The Actiheart device provides Inter-beat Intervals (R-peak intervals) extracted from processed ECG signals. We used the 5-minute time windows of the Inter-beat Interval (IBI) signal to calculate the heart rate (HR, heartbeats per minute), time-domain parameters (i.e., RMSSD, AVNN, SDNN, and pNN50), and frequency-domain parameters (i.e., low frequency (LF), high frequency (HF), and LF/HF). The HRV parameters are briefly described in Table 1. Further details about the heart rate and HRV features calculation have been previously described [4]. The abnormal IBI and HRV values generated by motion artifacts were removed before proceeding with the analysis according to the removal criteria described in another study utilizing HRV measures [28]. The removal criteria are based on the normal range of HR and IBI values.

Since the frequency of cortisol measurement was less than the measurement of the HRV parameters, we matched the cortisol value measured at the beginning of a 15-minute interval to the HRV parameter values measured at the three subsequent 5-minute time intervals. For instance, we matched the three 5-minute windows of HRV data between the 15- and

**Table 1. A summary of HRV parameters.**

| Parameter | Units | Description |
|-----------|-------|-------------|
| IBI | ms | Inter-beat interval; the time interval between two successive heartbeats |
| RMSSD | ms | The root mean square of successive differences between adjacent NN intervals |
| AVNN | ms | The average value of NN intervals |
| SDNN | ms | The standard deviation of normal NN intervals |
| pNN50 | - | The proportion of the number of pairs of successive NN intervals differing more than 50 ms divided by the total number of NN intervals |
| LF | ms$^2$ | Power of the low-frequency band of the IBI signal (i.e., 0.04 Hz—0.15 Hz) |
| HF | ms$^2$ | Power of the high-frequency band of the IBI signal (i.e., 0.15 Hz—0.4 Hz) |
| LF/HF | - | The ratio of LF to HF |

ms, milliseconds; NN, time (normalized) between two detected heartbeat detections.

30-minute saliva collection time points to a single cortisol value (i.e., the value measured at 15 minutes). We characterized the acute stress exposure period during visit 2 as the 30-minute window starting from the initiation of the TSST (i.e., between the 15–45 minute saliva collection time points). Data collected at all time points during visit 1 were considered for the non-stress status.

**Statistical analysis.** We first visually compared the pattern of HR, HRV features, and cortisol values across time between two visits for each participant using line graphs, and the average values for each measure at each visit per participant using bar graphs. The area-under-the-curve (AUC) values of HR, each HRV feature, and cortisol for the two study visits were calculated, using the formula shown below (1), where C(t) represents the value HR or HRV or cortisol values in time t.

$$AUC = \int_0^\infty C(t)dt \tag{1}$$

We used the paired-samples t-test to assess differences in the mean AUC values between visits using a p-value of <0.05 to indicate significance. We used Python and Python libraries, including Scipy [29] and Panda [30] to program the statistical analysis functions.

**Feature selection.** To maintain generalizability and avoid overfitting during the objective stress assessment, we implemented a feature selection method. Feature selection reduces the computational complexity, training time, and overfitting degree. Moreover, it improves the accuracy of the classification. Feature selection methods can be summarized into three categories: filter-based methods, wrapper-based methods, and embedded methods. We used a filter-based feature selection method which determines the relationship between multiple input features and target labels statistically. In other words, it can evaluate and filter out features that will be used in the classification model. Compared to the other two methods, the filter-based method is computationally cheaper and has less risk of overfitting [31].

In our filter-based methods, Gini impurity gain is added to find the most informative features for the assessment model. A decision-tree-based random forest classifier (max depth = 16) [32] is used to output the feature importance vector. Inside the decision tree model, each node is a condition on one of the features, and these nodes separate the data into two different sets. Data with the same labels are separated into the same set in an optimal scenario. The splitting condition depends on the impurity of the features chosen in each node. During the training process, the contribution to the decrease in the impurity of each feature is computed. Finally, the importance of each feature is ranked according to this measurement.

**Machine learning-based predictive models for stress assessment.** We developed a set of machine learning-based algorithms to detect the presence of acute stress exposure among pregnant women using HRV parameters and salivary cortisol. This is a binary classification between stress versus non-stress. Five different classification methods were implemented, including AdaBoost [33], XGBoost [34], Random Forest [32], Support Vector Machine (SVM) [35], and K-nearest-neighbor (KNN) classifiers [36]. The AdaBoost classifier is a meta-estimator that begins by fitting a classifier on the original dataset and then fits additional copies of the classifier on the same dataset but where the weights of incorrectly classified instances are adjusted such that subsequent classifiers focus more on difficult cases. XGBoost classifier is an optimized distributed gradient boosting library designed to be highly efficient, flexible, and portable. Random Forest classifier is an ensemble learning algorithm that fits a number of decision tree classifiers on various sub-samples of the dataset and uses averaging to improve the predictive accuracy and control over-fitting. Support Vector Machine is a margin-based classification technique used for the classification of linear as well as non-linear data. K-nearest-neighbor method uses k number of nearest data points and predicts the result based on a majority vote. We used the Scikit-learn library to create our classification models [37].

To accurately evaluate the performance of our classification models in terms of generalizability, the 10 folds Leave-one-subject-out cross-validation method [38] was used. Cross-validation is one of the most reliable algorithms that is used to objectively estimate the accuracy of a machine learning model on unseen data. It achieves this by training a model using different subsets of data and obtaining the average accuracy on the rest of the data as a test. In our study, among each iteration of our Leave-one-subject-out cross-validation method, 28 out of the total 29 participants' data were included for model training. For testing, only the remaining participants' data points were used. The final assessment accuracy of the model was obtained by averaging the accuracy of all constructed models.

## Results

Thirty-three participants completed the study and of these, HRV and cortisol data at both study visits were available for 29 participants. The mean PSS score was 13.4±3.7 out of a potential range of 0–40. PSS values from 14–26 indicate moderate stress and values from 27–40 indicate high stress levels. Thus, on average, this population of pregnant women had borderline moderate levels of perceived stress in their daily lives.

### Within-subject comparisons of parameters

We first visually compared the HR, HRV parameters, and cortisol in 5-minute time windows separately for each participant across the two studies visits. The comparison results can be summarized into three distinct types, representing inter-individual variability in the physiological response to acute stress exposure. Seventeen participants display an obvious difference in HR, HRV parameters, and cortisol between the stress and non-stress conditions. One example of this type of response is shown in Fig 2. Eight participants do not display a clear difference in HR and HRV parameters but do display a notable difference in cortisol, as shown in Fig 3. In the remaining 4 participants, no obvious difference in any of the parameters including cortisol could be detected. Fig 4 presents an example of this pattern of physiological non-responsiveness to the stress exposure. This variability in stress reactivity creates challenges for standard statistical techniques to detect the presence of stress.

We also visually compared the average value for all HRV features and cortisol between study visits for each participant (Fig 5). Among the 29 participants, we see that 19 participants have higher heart rate, 12 participants have higher RMSSD, 17 participants have higher

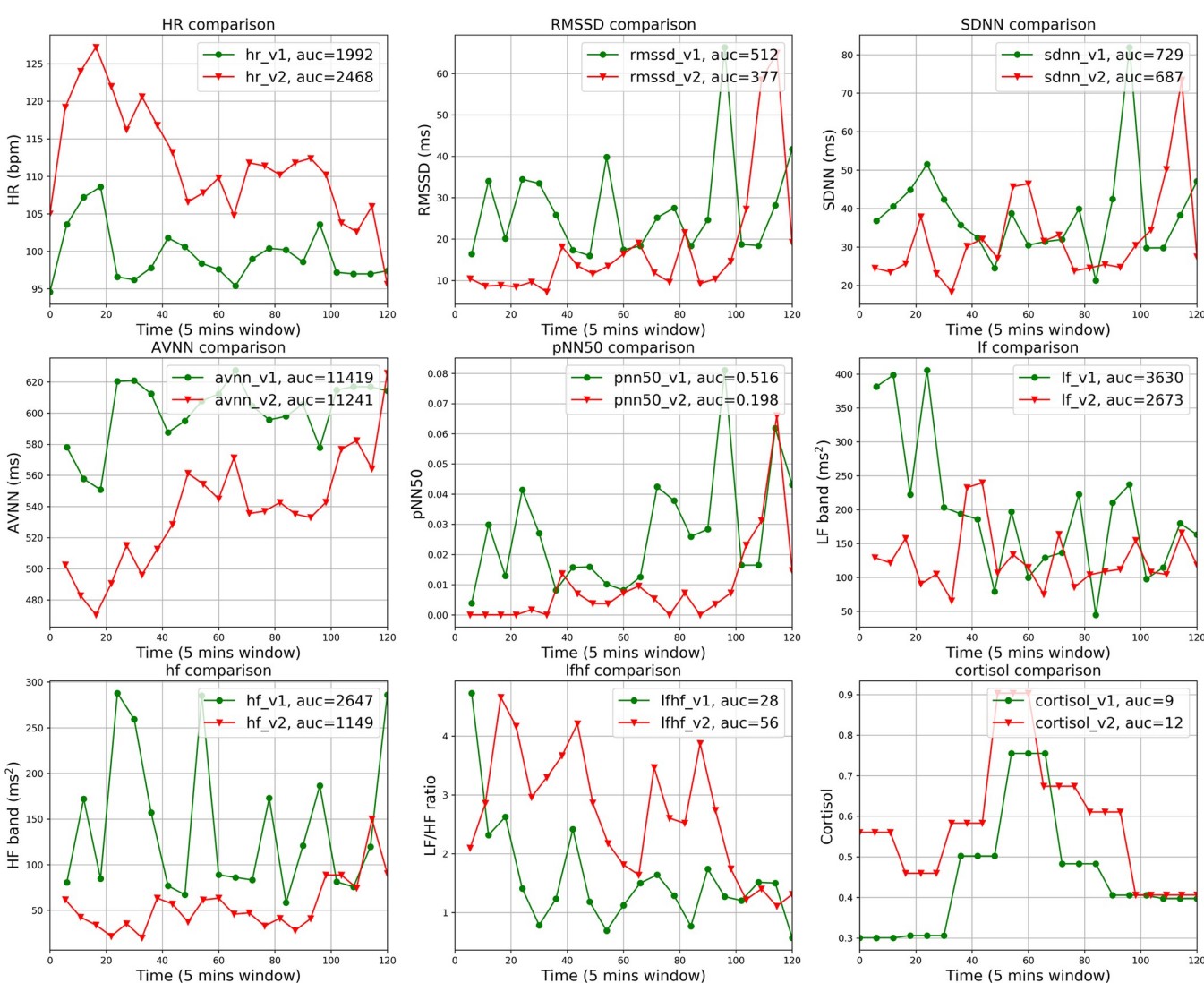

**Fig 2. Comparison of the HR, HRV parameters, and cortisol between visit 1 and visit 2 in 5-minute segments for a sample participant.** This participant demonstrated an obvious difference in HR, HRV parameters, and cortisol.

SDNN, 13 participants have higher AVNN, and 12 participants have higher pNN50 on exposure to stress (visit 2) versus non-stress (visit 1) conditions. As for frequency-domain features, 14 participants have higher LF band, 17 participants have higher HF band, and 12 participants have higher LF/HF ratio in visit 2 compared to visit 1. Finally, 23 participants have a higher average cortisol value in visit 2 compared to visit 1, which is the most obvious feature. In summary, for most participants, average values for HR, SDNN, HF, LF/HF ratio, and cortisol tend to be higher under acute stress, while other time- and frequency-domain parameters tend to be lower compared to the non-stress condition upon visual inspection.

Results of the paired t-test comparing the group mean AUC data for 9 features between visits are summarized in Table 2. Only the AUC for heart rate is statistically significantly different between visits, such that on average, participants had higher heart rate throughout the stress visit versus the non-stress visit.

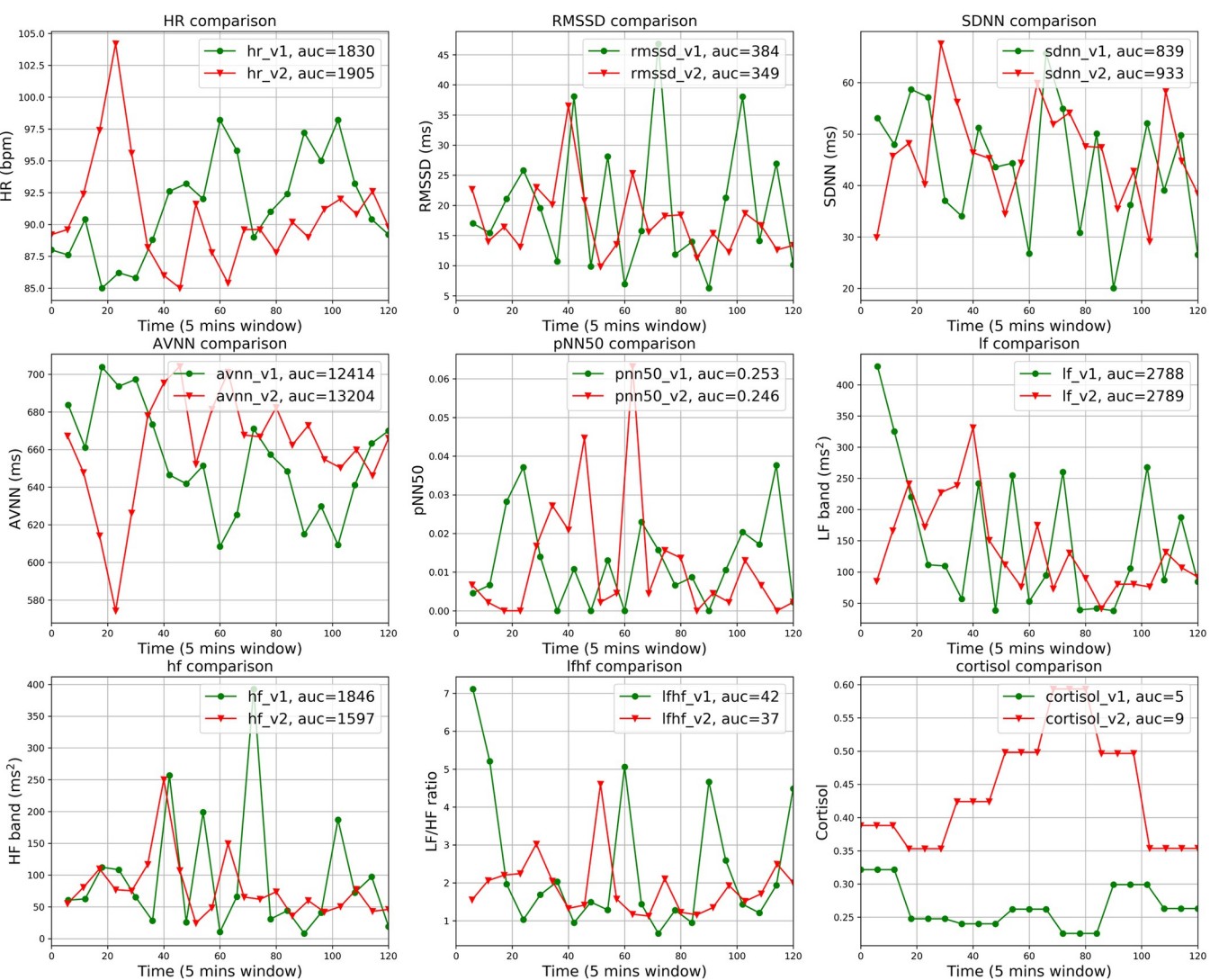

**Fig 3. Comparison of the HR, HRV parameters, and cortisol between visit 1 and visit 2 in 5-minute segments for a sample participant.** This participant did not demonstrate a clear difference in HR and HRV parameters but did have a significant difference in salivary cortisol.

**Objective stress assessment results.** We built an objective stress assessment model using the 9 features mentioned above based on machine learning methods. In order to reduce the computational complexity and save training time, we implemented a feature selection method using Gini impurity gain.

We first conducted the feature selection on HR and HRV parameters only, excluding cortisol. According to the feature selection results, 5 out of 8 features were selected as the best combination. The selected features in the order of importance are AVNN, HR, SDNN, LF, and LF/HF, indicating that AVNN is the most important feature for accurately detecting stress reactivity in the absence of cortisol. HR represents the frequency of heart beats and AVNN represents the average length of heartbeat intervals. Although AVNN and HR appear highly related from a statistical perspective, they represent distinct HRV features and thus, both contribute value to the model. When cortisol is added into the feature selection process, the order of importance among the features is cortisol, SDNN, AVNN, LF/HF, pNN50, and LF. Cortisol makes

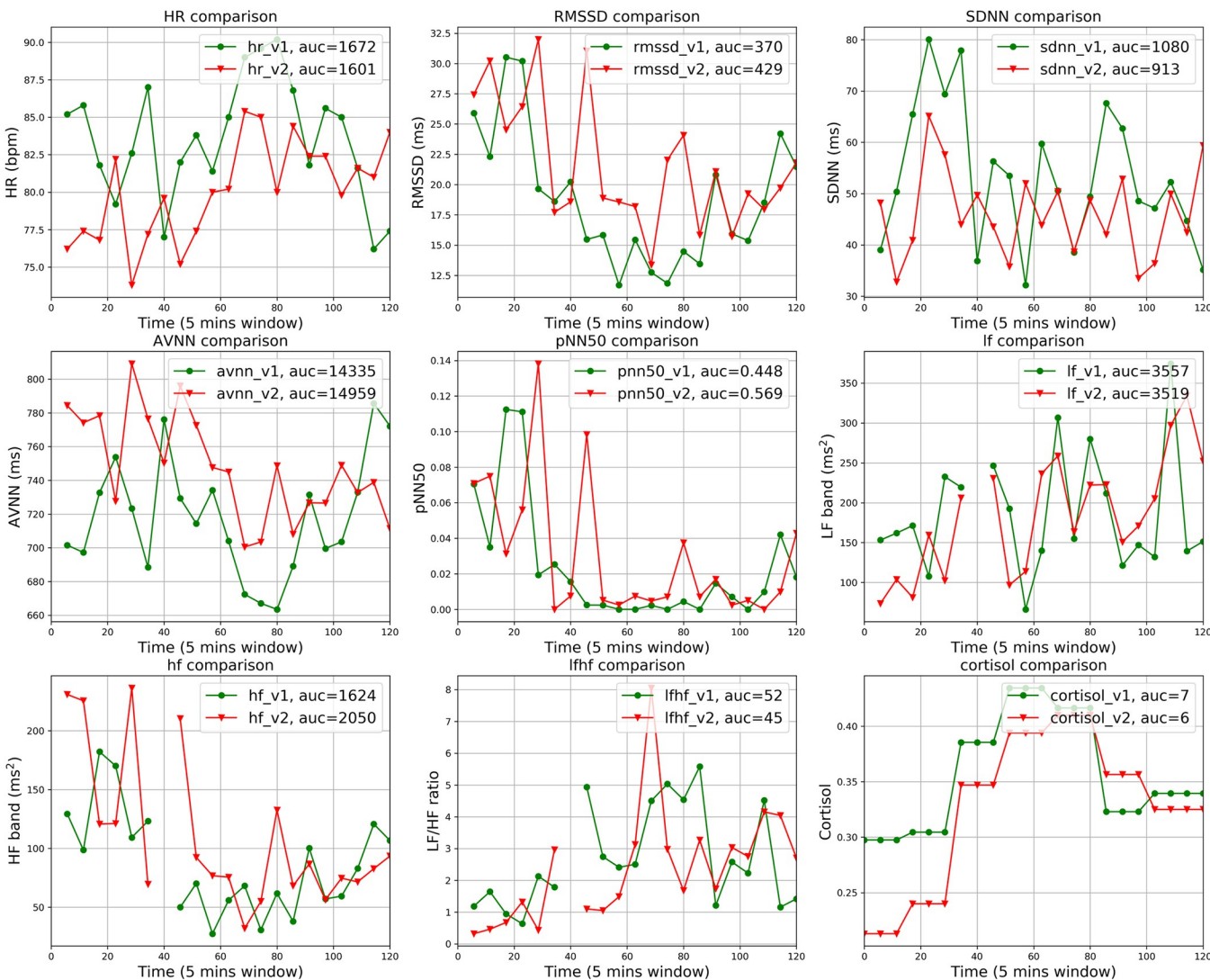

**Fig 4. Comparison of the HR, HRV parameters, and cortisol between visit 1 and visit 2 in 5-minute segments for a sample participant.** This participant did not demonstrate any clear difference among all the features, including salivary cortisol.

the largest contribution to the accuracy of stress detection. Among all the HRV features, SDNN contributes most to the assessment accuracy in the presence of cortisol, and HR is no longer selected as an important feature.

Following feature selection, we built the stress assessment model using the top 5 features selected in the absence of cortisol using 5 different machine learning models. The assessment results are shown in Table 3. The AdaBoost classifier has the best performance with an assessment accuracy of 67.11%. The model was then repeated using the 6 features selected from 9 parameters including cortisol. AdaBoost remained the best classifier with an increased assessment accuracy of 76.51%, which is a 9.4% improvement compared to the stress assessment model without cortisol.

The performance of the AdaBoost classifier with six selected features using the Leave-one-subject-out cross-validation method is shown in Fig 6. The highest accuracy is 92.31% and the lowest accuracy is 42.51%. For other classifiers, the accuracy improvement after adding cortisol

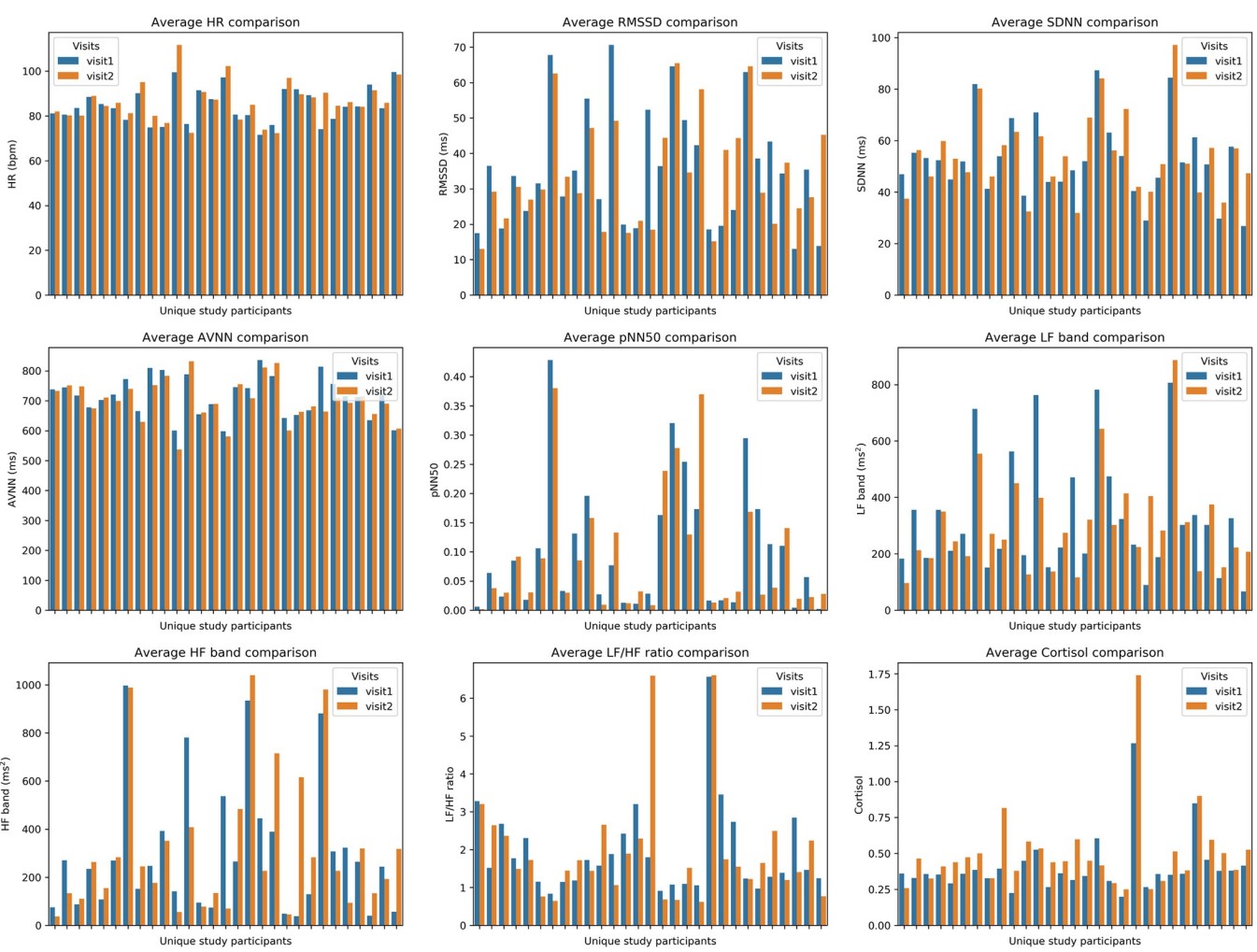

**Fig 5. Comparison of average values for HR, HRV parameters, and cortisol between visit 1 and visit 2 for all 29 participants.**

as a feature is 8.67% for the XGB classifier, 9.51% for the Random Forest classifier, 10.51% for the SVM classifier, and 13.12% for the KNN classifier. That is an average 10.24% increase after introducing cortisol into the stress assessment model. The stress assessment results using all 9

**Table 2. Within-subject comparison of AUC values for HR, HRV parameters, and cortisol between stress and non-stress conditions.**

| Features | Visit 1, non-stress condition (Mean±SD) | Visit 2, stress condition (Mean±SD) | t value | df | P-value |
|---|---|---|---|---|---|
| HR | 1726.31 ± 209.03 | 1836.13 ± 263.57 | -3.285 | 28 | 0.003 |
| RMSSD | 649.13 ± 309.35 | 633.53 ± 285.59 | 0.369 | 28 | 0.715 |
| SDNN | 971.17 ± 289.14 | 1015.08 ± 317.29 | -1.276 | 28 | 0.213 |
| AVNN | 13195.91 ± 2156.73 | 13219.98 ± 2540.54 | -0.149 | 28 | 0.883 |
| pNN50 | 1.91 ± 2.10 | 1.76 ± 2.10 | 0.634 | 28 | 0.531 |
| LF | 5606.18 ± 3400.39 | 5352.93 ± 3115.48 | 0.586 | 28 | 0.562 |
| HF | 5268.97 ± 4880.39 | 5523.71 ± 5213.20 | -0.412 | 28 | 0.683 |
| LF/HF | 34.05 ± 23.96 | 34.98 ± 26.93 | -0.262 | 28 | 0.796 |
| Cortisol (standardized for time of awakening) | 0.07 ± 1.05 | 0.04 ± 1.00 | 0.207 | 28 | 0.837 |

**Table 3. Stress assessment results.**

|  | AdaBoost | XGBoost | Random Forest | Support Vector Machine | K Nearest Neighbor |
|---|---|---|---|---|---|
| 5 selected features in the absence of cortisol | 67.11% | 65.43% | 64.04% | 64.14% | 60.15% |
| 6 selected features in the presence of cortisol | 76.51% | 74.1% | 73.55% | 74.65% | 73.27%% |
| All 9 features (including cortisol) | 73.09% | 72.21% | 72.08% | 73.1% | 69.5% |

features without the feature selection are also shown in Table 3. There is a slight accuracy drop for all the classifiers besides the increase in computational complexity and training time.

## Discussion

### Principle results

This study demonstrates wide inter-individual variability in bio signal responses to acute psychological stress, including HR, HRV, and salivary cortisol values, in a cohort of pregnant women. Despite this variability, using a machine learning-based algorithm, we identified the optimal combination and classification of stress bio signals that are capable of assessing the presence of stress with almost 77% accuracy. Salivary cortisol contributed a significant increase in accuracy of the assessment model compared to using a range of HRV parameters without cortisol.

The fluctuation of HR, time-domain HRV, and frequency-domain HRV parameters across stress and non-stress conditions was evident for more than half of the study participants (17 out of 29), which suggests that these parameters in isolation may only be moderately reliable signs of stress exposure. In comparison, a noticeable visual difference in cortisol values across the two visits was observed for 25 out of 29 participants, although on average, the difference in cortisol AUC was not statistically significant between the stress versus non-stress visits. This

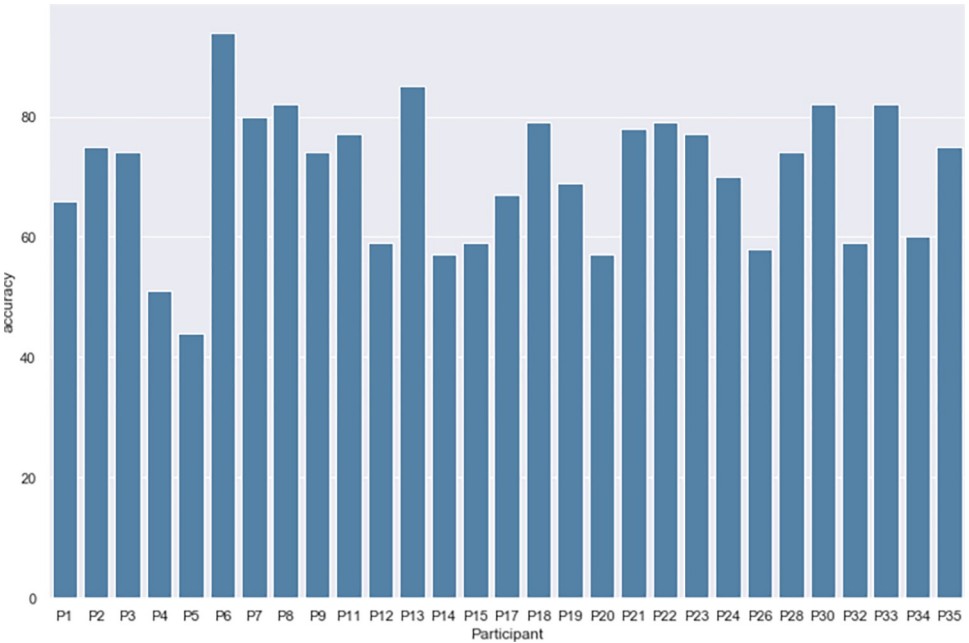

**Fig 6. The performance of the AdaBoost classifier with 6 selected features using the Leave-one-subject-out cross-validation method.**

points to the potential unreliability of using standard statistical methods and salivary cortisol alone as an objective measure of acute stress. Indeed, salivary cortisol is a more sensitive measure of HPA-axis function than serum cortisol since it is present only in the unbound, biologically active form in saliva [39]. Cortisol binding proteins in plasma increase markedly during pregnancy thereby reducing concentrations of the active hormone. Although both HRV and salivary cortisol levels reflect ANS activation, HRV parameters are affected by the sympathetic adrenal medullary (SAM) axis rather than the HPA-axis which triggers cortisol release. These different modes of activation may contribute to the observed differences in fluctuations of these bio signals under stress.

HR and HRV parameters can be measured easily using a chest band, a smartwatch, or a ring [28] in everyday life or in laboratory settings making it a feasible approach for most research studies. However, our results suggest that among pregnant women, HRV parameters alone may not be adequate as an acute stress assessment tool but may benefit from other bio signals to help distinguish between stress and non-stress states more accurately. Furthermore, ECG signals may be influenced by factors other than psychological stress. The results of the paired t-test in our study indicated a significant difference in HR between the stress and non-stress conditions. Besides the stimulation of stress, increased HR on the stress visit may also be related to the fact that women walked a short distance to a separate room for the TSST, while on the non-stress visit, they remained in the same room but stood up during the task period. These additional movements may have increased their HR to some extent around the time of stress exposure. Meanwhile, no significant difference was found in HRV parameters suggesting that standard statistical methods may not be sufficient to detect average HRV differences across a group. It is also important to note that 4 participants failed to demonstrate any notable difference in salivary cortisol, HR, or HRV between the stress and non-stress conditions. More advanced analytical approaches, such as machine learning-based feature selection, may be required in such cases to optimize the data and distinguish between stress and non-stress situations.

We implemented the machine learning-based feature selection using the Gini impurity index to reduce the computational complexity and avoid overfitting. Moreover, we present the most important features of the objective stress assessment model when cortisol is added by repeating the feature selection. The feature selection results differ from the statistical analysis because this method makes decisions according to the contribution of stress assessment accuracy. If the accuracy is not improved when a feature is added, these features will not be used further in the assessment model.

We also tested the performance of our model to reliably assess stress using five different machine learning methods. The assessment accuracy increased on average 10% across all five classifiers after adding cortisol as a feature. The significant difference in cortisol values for most participants helps the model to better distinguish between stress and non-stress states. Because of participants' different HR, HRV, and cortisol reactions to stress exposure, the best performance model, AdaBoost classifier, achieved assessment accuracy higher than 90% and lower than 30% with different participants by using the Leave-one-subject-out cross-validation method. When we use all nine features, the assessment accuracy decreases slightly. The remaining three features add more computational complexity, resulting in model overfitting in the training process, and a poor performance during testing. The AdaBoost classifier achieved the highest assessment accuracy of almost 77% with the selected six features. These results demonstrate that our algorithm can assess an acute stress state for pregnant women using a combination of HRV parameters and salivary cortisol values with acceptable accuracy.

## Comparison with previous studies

To the best of our knowledge, this is the first study to develop a machine learning-based algorithm to detect the presence of acute stress exposure among pregnant women using a combination of HRV features and salivary cortisol values. Ilyumzhinova et al. [40] reported the association between changes in salivary cortisol and RMSSD (one of the time-domain HRV parameters) among black pregnant women exposed to the TSST. They found that higher self-reported levels of stress related to experiences of discrimination among participants were associated with lower levels of cortisol reactivity and higher levels of RMSSD following the TSST, which demonstrates how baseline maternal factors may influence an individual's stress responsivity. However, this study did not evaluate other HRV parameters in response to the TSST, nor did they utilize a combination of cortisol and HRV features to characterize the presence or absence of stress.

Previous studies that have attempted to objectively characterize stress in healthy, non-pregnant subjects report varying degrees of success. Bakker and Pechenizkiy proposed a stress assessment system using galvanic skin response (GSR), a measure of skin conductance that is affected by sweat gland activity under stress [41]. The authors concluded that using this bio signal in isolation was insufficient to determine the presence of stress with high accuracy. Sun et al. [42] used a physical activity protocol to induce stress and collected ECG, GSR, and accelerometer data for 30 minutes to implement a stress detection system. This study detects mental stress affected by physical activities. Han et al. [43] assessed stress among healthy people using 25 features extracted from ECG, Photoplethysmograph (PPG), and GSR signals. They achieved an accuracy similar to ours at the cost of huge computational complexity. In another study [44], a stress detection tool was developed using ECG, GSR, respiration rate, blood pressure, and peripheral capillary oxygen saturation (SpO2) using a waist strap. Although this study included multiple bio signals, the highest accuracy achieved by the Leave-one-subject-out cross-validation method was 89.3%, which is lower than the performance of our model (93.55%). None of these studies have pregnant participants, and their methods were all focused on healthy subjects. We identified only one published study that built a stress detection system for pregnant women [45]. This study implemented the TSST to induce stress and collected HR, ECG, GSR signals during the experiment. With more than 20 features extracted from these three signals, they achieved an average stress assessment accuracy of 70%, and their best Leave-one-subject-out accuracy was 81%. Our model performed better in both measurements using only six features that included salivary cortisol.

Future research is required to increase the accuracy of stress assessment by incorporating more bio signals such as GSR in combination with cortisol and HRV and applying more advanced machine learning methods. Although saliva samples can be conveniently collected by a cotton swab in free-living and human laboratory settings, this method is more invasive for participants than wearable sensors and performing cortisol assays adds a level of complexity.

## Strengths and limitations

The crossover design is a strength of this study as it facilitates intra-individual comparison of physiological responses to stressful and non-stressful tasks. Also, the strict eligibility criteria concerning the gestational age at study assessments, ethnicity, and pre-pregnancy BMI range help to minimize inter-individual variation and thereby increase the validity of the results. We chose to study a cohort of Hispanic pregnant women as the Hispanic population in the U.S. may be subject to higher levels of baseline stress than non-Hispanic people due to various sociodemographic and acculturative factors. While our cohort perceived borderline moderate

levels of stress over the past month, it is unknown whether such baseline stressors contribute to higher or lower stress responses to the controlled stress task compared to non-Hispanic women. Furthermore, our findings cannot necessarily be generalized to other racial/ethnic groups, or to non-pregnant or free-living subjects encountering everyday stressors. However, data collection in free-living situations may be subject to more motion artifact noise in physiological signals in comparison with data collection in a controlled environment and mostly stationary state (especially for ECG data which is worn on the participant's chest).

We also acknowledge that the use of a laboratory-based standardized stress protocol does not necessarily reflect real-world stressful events, and indeed not all participants in this study demonstrated a physiological stress response to the task. However, the TSST is the most widely used, validated protocol for stress research in laboratory settings and the machine learning approach helps to overcome the issue of non-responders to the stressor by utilizing those data to train the model with greater accuracy to detect the presence of stress. Introduction of meal ingestion as part of the parent study could also be considered a limitation for the present analysis, as it is unknown to what degree the act of consuming a meal may have moderated the stress response. However, as the meal type and time of ingestion was standardized across all subjects and occurred on both visits, we assume that any potential influence of the meal on the physiological stress response is also standardized across participants. While we adjusted cortisol values for time of awakening at each visit, it is also possible that stressful experiences occurred for some participants before arriving to their study visits, which could have elevated their baseline cortisol values. We were unable to account for these potential sources of external stress in the data.

Other limitations of this study relate to the data processing procedures. The Acticheart only provided the IBI data and not the raw ECG signal, which may have facilitated the calculation of more accurate HRV parameters. Removal of corrupted IBI data due to motion artifacts resulted in some gaps in our HRV feature comparisons. Most of these gaps occurred at the very beginning of data collection, in the 5 minutes before participants underwent the stress or non-stress task period. This resulted in inadequate baseline HRV measurements from which to compute reactivity scores for each parameter, which may have been a more robust summary measure than the AUC values on which we relied for the statistical analysis. As noted above, salivary cortisol values were only available at set intervals according to the timing of sample collections, while HRV parameters are continuously measured and data reported in 5-minute intervals. As a result, the cortisol value in three consecutive 5-minute segments was assumed to be the same for the purpose of the feature selection models. If it would be possible to measure cortisol every 5 minutes, there is potential to further increase the accuracy of our stress assessment model.

## Conclusion

In summary, our results demonstrate wide variability in objectively measured stress reactivity signals among pregnant women exposed to a standardized laboratory stress task. HR, SDNN, HF, LF/HF ratio, and cortisol tend to be higher under acute stress, while other time- and frequency-domain parameters tend to be lower. By adding cortisol as a feature alongside HR and HRV parameters, our model can detect pregnant women's stress status with acceptable accuracy. Therefore, our objective stress assessment algorithm provides a framework for future clinical research studies to consider a combination of physiological stress signals and machine learning-based approaches to assess stress more accurately in human subjects. The potential translation of this work to the assessment of chronic stress in humans requires further research.

## Author Contributions

**Conceptualization:** Rui Cao, Karen L. Lindsay.

**Data curation:** Karen L. Lindsay.

**Formal analysis:** Rui Cao, Karen L. Lindsay.

**Funding acquisition:** Karen L. Lindsay.

**Investigation:** Karen L. Lindsay.

**Methodology:** Rui Cao, Amir M. Rahmani, Karen L. Lindsay.

**Project administration:** Karen L. Lindsay.

**Resources:** Karen L. Lindsay.

**Software:** Rui Cao.

**Supervision:** Amir M. Rahmani, Karen L. Lindsay.

**Validation:** Rui Cao, Karen L. Lindsay.

**Visualization:** Rui Cao, Karen L. Lindsay.

**Writing – original draft:** Rui Cao, Amir M. Rahmani, Karen L. Lindsay.

**Writing – review & editing:** Rui Cao, Amir M. Rahmani, Karen L. Lindsay.

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
