## [Decision Letter · Decision Letter 0]

30 May 2022

PONE-D-22-11525Prenatal Stress Assessment using Heart Rate Variability and Salivary Cortisol: A Machine Learning-Based ApproachPLOS ONE

Dear Dr. Lindsay,

Thank you for submitting your manuscript to PLOS ONE. After careful consideration, we feel that it has merit but does not fully meet PLOS ONE’s publication criteria as it currently stands. Therefore, we invite you to submit a revised version of the manuscript that addresses the points raised during the review process.

The reviewers have raised some issues that need to be addressed before the manuscript might recommended for publication.  Please submit your revised manuscript by Jul 14 2022 11:59PM. If you will need more time than this to complete your revisions, please reply to this message or contact the journal office at plosone@plos.org. Please include the following items when submitting your revised manuscript:A rebuttal letter that responds to each point raised by the academic editor and reviewer(s). You should upload this letter as a separate file labeled 'Response to Reviewers'.A marked-up copy of your manuscript that highlights changes made to the original version. You should upload this as a separate file labeled 'Revised Manuscript with Track Changes'.An unmarked version of your revised paper without tracked changes. You should upload this as a separate file labeled 'Manuscript'.If applicable, we recommend that you deposit your laboratory protocols in protocols.io to enhance the reproducibility of your results. Protocols.io assigns your protocol its own identifier (DOI) so that it can be cited independently in the future. For instructions see: https://journals.plos.org/plosone/s/submission-guidelines#loc-laboratory-protocols. Additionally, PLOS ONE offers an option for publishing peer-reviewed Lab Protocol articles, which describe protocols hosted on protocols.io. Read more information on sharing protocols at https://plos.org/protocols?utm_medium=editorial-email&utm_source=authorletters&utm_campaign=protocols.

We look forward to receiving your revised manuscript.

Kind regards,

Enzo Pasquale Scilingo, Ph.D.

Academic Editor

PLOS ONE

Journal Requirements:

Reviewers' comments:

Reviewer's Responses to Questions

**Comments to the Author**

1. Is the manuscript technically sound, and do the data support the conclusions?

Reviewer #1: Yes

Reviewer #2: Partly

2. Has the statistical analysis been performed appropriately and rigorously? 

Reviewer #1: Yes

Reviewer #2: I Don't Know

3. Have the authors made all data underlying the findings in their manuscript fully available?

Reviewer #1: No

Reviewer #2: Yes

4. Is the manuscript presented in an intelligible fashion and written in standard English?

Reviewer #1: Yes

Reviewer #2: Yes

5. Review Comments to the Author

Reviewer #1: This is an interesting topic and the authors have done a good job with the manuscript. However, my main concern is in regard to the rationale behind the study. After reading the manuscript I’m still not sure what the benefit is of undertaking machine learning to identify stress in this population and how this might be practically useful.

A few comments:

Abstract

• From reading the aims and methods sections, I’m not actually clear what data the machine learning is based on.

• I’m also not clear on the importance of this research – could the objective be a little clearer in terms of application / background

Introduction

• Some of the language is a little informal e.g., ‘wear and tear’, ‘people nowadays’ etc.

• I’m not sure what a stress pulse survey is. Is this scientific?

• The stressors described on like 76 seem to have been chosen at random. Surely racial discrimination is not pregnancy specific? Also what is the difference between perceived stress and stress?

• I don’t buy the argument that stress must be assessed during pregnancy. Why is it important to measure physiological stress rather than via self-report? Surely the practicality of self-report in a medical context outweighs the drawbacks, and is it likely that cortisol and/or HRV will be measured in a clinical context? This section could be stronger.

Methods

• Why were participants excluded if they were not of Hispanic ethnicity? This is a limitation in terms of generalizability

Results

• Figure 2 – 4 are very repetitive. It may be more useful to present data on the whole rather than for individual participants.

• There are a number of statements about variables being ‘higher’ or ‘lower’ under certain conditions. On what basis are these judgements made? It does not seem that any statistical or numerical data is presented.

• Information missing for t test data (e.g., t value, df)

Discussion

• Is it possible that some of the participants simply didn’t find the test stressful? This may explain why HR fluctuations etc were not seen

• Given the reliability of cortisol as a measure of stress as discussed, I’m still unclear on the aim of this study. Why would we need to use machine learning to identify stress using cortisol outcomes? Why not just use the cortisol outcomes themselves?

• Unclear what is meant by ‘discrimination stress’ line 403. Was this in the study referred to?

• Line 445 – was baseline stress measured in the chosen population? If not I’m not sure this argument is very strong.

Reviewer #2: 1 Technical soundness:

The medical results of this experiment are limited to pregnant women of Hispanic origin who were overweight before pregnancy. There seem to be minor flaws in the collection of data, as described below. The attempt to build an objective stress assessment model is interesting and commendable, with an acceptable accuracy (average 77%), although visual assessment of graphs classified 86% of cases correctly. Medically, it is more interesting to objectively detect and evaluate chronic stress exposure rather than acute stress.

This is a secondary analysis of a cross-over study which primarily intended to study the effect of psychological stress on postprandial metabolic response to a standardized meal. The metabolic data are not accounted for in the paper. Supposedly they are subject for a different publication.

Strictly speaking, therefore, the intervention in this experiment is primarily the ingestion of a standardized meal (which normally triggers the parasympathetic autonomic system) and, secondly, a stress test task (which activates the sympathetic autonomic system). The superimposed triggers of the autonomic nervous system of the experiment complicate the interpretation of data and might explain some of the heterogeneity noted. The authors might elaborate on the effect of the dual intervention in their text.

The experiments in the study seem to have been conducted rigorously, except for the fact that visit 2 (TSST visit), other than introducing the intended psychological stress, inadvertently added physical movements (the women had to walk to a separate room) which might have increased heart rate and therefore influenced also the HRV parameters. The authors comment appropriately on this in the discussion section (line 368).

Another concern, which is not commented on, is that the time of awakening of the participants are missing and might have been different on the two occasions. Time of awakening is crucial to the assessment of cortisol levels, since the time from the morning peak values of cortisol would have influenced the base line cortisol values, and might not be comparable between the two occasions (although the participants arrived at the research centre at about the same time). If most of them rose later in the morning on visit 2, for example, they would indeed display higher cortisol levels from start. It would therefore probably have been more correct to analyse changes from baseline of cortisol rather than absolute cortisol levels and AUC. The level of stress in the morning in the participants’ everyday home environment, before arriving at the research centre, are not taken into account and might also be of importance for base line cortisol levels.

A technical weakness, which the authors already point out in the text (line 454), is that the Actiheart registration did not provide a raw ECG-signal, which risks to make the HRV data less accurate.

Line 202: Based on what criteria were Actiheart artifacts removed (since no visual inspection of ECG was possible)?

2 Statistics

The time t used for calculation of the AUC seems to be the same for cortisol and for HRV-parameters (namely 120 minutes). This probably suits cortisol dynamics which are quite slow, but does not match HRV dynamics, since parasympathetic withdrawal and reinstatement are almost immediate in effect, and the researchers will therefore in the same laps of time capture instances of stress (with lower HRV), but possibly also a period of relaxation/ relief after stress, which might translate into higher HRV, thus levelling out the AUC on visit 2 (particularly of the parasympathic indices). This might explain why the researchers found no statistical difference in HRV parameters when comparing AUCs. It probably does not affect the objective stress assessment model, however.

This reviewer has no deeper insight in programming or in machine learning-based algorithms and can therefore only comment on the quality of medical data used for feature selection. To this reviewer’s knowledge all HRV parameters are influenced by HR, are highly interdependent and are not normally distributed. HR and AVNN are inherently redundant - it is surprising the feature selection allows both, considering it is constructed to avoid overfitting?

3 All data supporting the conclusions seem to be fully available in the Supporting information file. (This is a secondary analysis of a cross-over study which primarily intended to study the effect of psychological stress on postprandial metabolic response to a standardized meal. The metabolic data are not accounted for in the paper. Supposedly they are subject for a different publication.)

4 This work is presented in intelligible standard English.

Line 108: Statement should be backed up by a reference.

Line 172 and 178: TSST reference?

Line 216: “,” instead of “.” as in one sentence.

Line 453: Actiheart instead of Actiheat.

Figure 5: Dots should not be connected by lines, and it should be clarified that unlike Figure 2, 3 and 4, x is no longer time, but represent discrete individuals.

6. PLOS authors have the option to publish the peer review history of their article (what does this mean?). If published, this will include your full peer review and any attached files.

Reviewer #1: No

Reviewer #2: No

---

## [Author Response · Author response to Decision Letter 0]

28 Jun 2022

We thank the reviewers for their thorough review of this manuscript and helpful feedback. We have addressed each comment below and annotated the relevant changes made to the manuscript. 

In response to the query of data availability from reviewer #1, we have uploaded the study data to an online repository which may be accessed at this link during the review process: https://datadryad.org/stash/share/8SKG9TIJOiVgeV90rKvNBI34O2x6eyNgjjgxt4Om8CE

REVIEWER #1: 

This is an interesting topic and the authors have done a good job with the manuscript. However, my main concern is in regard to the rationale behind the study. After reading the manuscript I’m still not sure what the benefit is of undertaking machine learning to identify stress in this population and how this might be practically useful.

A few comments:

Abstract

Comment #1: From reading the aims and methods sections, I’m not actually clear what data the machine learning is based on.

Response: We thank the reviewer for their comments. The machine learning algorithms use the physiological data collected from pregnant women including Heart Rate, Heart Rate Variability (RMSSD, AVNN, SDNN, pNN50, LF Band, HF Band, and LF/HF Ratio), and salivary cortisol. Salivary cortisol is considered the primary stress hormone but it is not always convenient to measure in free-living subjects. Meanwhile, heart rate variability (HRV) offers a convenient alternative measure of stress physiology using non-invasive wearable technology. However, there are limited studies that consider the reliability of HRV as an indicator of prenatal stress, and the combination of cortisol and HRV assessment in pregnancy has not yet been tested. Therefore, in this study, we applied machine learning methods to assess the presence of acute stress in pregnant women using these multimodal parameters simultaneously. 

We have updated the methods description in the abstract for greater clarity on the input features (line 28-30):

“A filter-based features selection method was used to identify the importance of different features [heart rate (HR), time and frequency-domain HRV parameters and salivary cortisol] for stress assessment and reduce the computational complexity.”

We have also added the following sentence to the introduction to justify the rationale for the machine learning approach (line 103-106):

“Statistical methods can measure stress according to one or two physiological features at a time. However, machine learning approaches can assess stress objectively by fusing multiple physiological features, and even multiple modalities, thereby increasing efficiency and accuracy.”

Comment #2: I’m also not clear on the importance of this research – could the objective be a little clearer in terms of application / background

Response: We have clarified the objective statement and its application to read as follows (line 23-25): 

“To develop a machine learning-based algorithm utilizing heart rate variability and salivary cortisol to detect the presence of acute stress among pregnant women that may be applied to future clinical research”

Introduction

Comment #3: Some of the language is a little informal e.g., ‘wear and tear’, ‘people nowadays’ etc.

Response: We have revised these sentences to read as follows (lines 66-67 and 69-70):

“...the growing mental burden from societal expectations and daily workload in the present day contributes to pervasive, chronic stress”

“Prolonged stress is usually associated with depletion of overall health…”

Comment #4: I’m not sure what a stress pulse survey is. Is this scientific?

Response: Thank you for the comment. The stress pulse survey was conducted by the ComPsych company. ComPsych is the pioneer and the world's largest provider of employee assistance programs, servicing more than 60,000 organizations and 130 million individuals throughout the U.S. and 190 countries. Therefore, the survey is a robust reflection of the global working population. We have removed the term “Stress Pulse” from the manuscript text and replaced it with the word “comprehensive” to avoid any confusion among readers (line 67-69).

Comment #5: The stressors described on line 76 seem to have been chosen at random. Surely racial discrimination is not pregnancy specific? Also what is the difference between perceived stress and stress? 

Response: Thank you for highlighting this valid point. We have rewritten the sentence to emphasize the additional stressors unique to pregnancy, as follows (line 75-78):

“Pregnant women may encounter various stressors that exceed those of everyday life for non-pregnant people including worries about the wellbeing of themselves and their developing baby, concerns about delivery, and financial or other socioeconomic stressors associated with a growing family.”

Comment #6: I don’t buy the argument that stress must be assessed during pregnancy. Why is it important to measure physiological stress rather than via self-report? Surely the practicality of self-report in a medical context outweighs the drawbacks, and is it likely that cortisol and/or HRV will be measured in a clinical context? This section could be stronger. 

Response: We appreciate the reviewer’s comment on this point. We have added the following sentences to the Introduction to justify the importance and applicability of objective stress measures in the context of pregnancy (line 93-100):

“In standard clinical practice with pregnant women, stress assessment surveys are not typically used due to time constraints and the various limitations of these methods as noted above. As research on objective stress characterization using real time, non-invasive technology evolves, there is potential for these techniques to be translated to clinical settings so that clinicians and/or their pregnant patients may benefit from momentary feedback on patients’ physiological state. In turn, this could facilitate implementation of simple, effective stress management techniques that could reduce the adverse effects of stress on health outcomes for mother and baby.”

Methods

Comment #7: Why were participants excluded if they were not of Hispanic ethnicity? This is a limitation in terms of generalizability

Response: We agree with the reviewer that the ethnicity criteria is a limitation on generalizability for this study, and this has been acknowledged in the limitations section of the discussion. The decision to restrict to Hispanic ethnicity was based on the parent study design and outcome measures. We have added the following sentences to the methods section to provide further clarity in this regard (line 162-166):

“The parent study restricted eligibility to women of Hispanic ethnicity in order to have a homogenous cohort of participants known to experience higher levels of socio-cultural stressors. Homogeneity among participants is important to limit inter-individual variability that could influence metabolic biomarkers that are primary outcome measures for the parent study.”

Results

Comment #8: Figure 2 – 4 are very repetitive. It may be more useful to present data on the whole rather than for individual participants.

Response: We appreciate this comment and while we understand that figures 2-4 might appear to present repetitive data, we wish to clarify the important differences and value that each figure holds for the reader. We have updated the text in this part of the results as follows (line 299-308):

“The comparison results can be summarized into three distinct types, representing inter-individual variability in the physiological response to acute stress exposure. Seventeen participants display an obvious difference in HR, HRV parameters, and cortisol between the stress and non-stress conditions. One example of this type of response is shown in Figure 2. Eight participants do not display a clear difference in HR and HRV parameters but do display a notable difference in cortisol, as shown in Figure 3. In the remaining 4 participants, no obvious difference in any of the parameters including cortisol could be detected. Figure 4 presents an example of this pattern of physiological non-responsiveness to the stress exposure. This variability in stress reactivity creates challenges for standard statistical techniques to detect the presence of stress.”

By displaying these three figures of distinct patterns in stress reactivity, we demonstrate to the readers that individuals tend to differ significantly in response to stress and these varying reactions have a significant impact on our algorithm’s stress assessment accuracy.

Comment #9: There are a number of statements about variables being ‘higher’ or ‘lower’ under certain conditions. On what basis are these judgements made? It does not seem that any statistical or numerical data is presented. 

Response: We understand that this comment refers to the results text that relates to Figure 5. These descriptive comparisons are based on visual inspection of differences in AUC values for each parameter, as distinct from the statistical testing of differences that is described in the following paragraph with Table 2. We have updated the manuscript accordingly to ensure that the reader is clear where we refer to higher or lower measures on visual inspection versus the subsequent statistical testing by paired t-tests (line 326-328):

“In summary, for most participants, average values for HR, SDNN, HF, LF/HF ratio, and cortisol tend to be higher under acute stress, while other time- and frequency-domain parameters tend to be lower compared to the non-stress condition upon visual inspection.”

Comment #10: Information missing for t test data (e.g., t value, df)

Response: These values have been added to Table 2 (page 18)

Discussion

Comment #11: Is it possible that some of the participants simply didn’t find the test stressful? This may explain why HR fluctuations etc were not seen

Response: We agree with the reviewer that it is likely that some participants did not find the TSST stressful and this is reflected in the visualization of HRV and cortisol features displayed in figure 4. In the updated cortisol AUC data presented in Table 2 (values normalized for time of awakening), we also do not see a statistically significant difference in total cortisol output on average between the stress and non-stress conditions. The value in the machine learning approach is that it makes use of this variation in stress responses across participants to train the model to better detect true stress exposure with greater accuracy than standard statistical methods. We have added the following to the limitations section to acknowledge this limitation but also the value it provides for this analysis (line 493-498):

“We also acknowledge that the use of a laboratory-based standardized stress protocol does not necessarily reflect real-world stressful events, and indeed not all participants in this study demonstrated a physiological stress response to the task. However, the TSST is the most widely used, validated protocol for stress research in laboratory settings and the machine learning approach helps to overcome the issue of non-responders to the stressor by utilizing those data to train the model with greater accuracy to detect the presence of stress.”

Comment #12: Given the reliability of cortisol as a measure of stress as discussed, I’m still unclear on the aim of this study. Why would we need to use machine learning to identify stress using cortisol outcomes? Why not just use the cortisol outcomes themselves?

Response: We appreciate this comment. On this resubmission, we wish to highlight that our revised data better supports the rationale to use HRV data in addition to cortisol for assessing the presence of stress. On recommendation of reviewer 2 (comment #4), we have generated normalized cortisol values that account for the influence of time of awakening at each cortisol measurement point. We re-computed the cortisol AUC using these updated values and note that on the paired t-test, we no longer see a statistically significant difference in cortisol AUC between the stress and non-stress visits (Table 2). Thus, we cannot say definitively that cortisol alone is sufficient to discriminate a stressful exposure, even in the controlled lab-based setting utilized in our study. We have further expanded on this point in the discussion (lines 386-390):

“In comparison, a noticeable visual difference in cortisol values across the two visits was observed for 25 out of 29 participants, although on average, the difference in cortisol AUC was not statistically significant between the stress versus non-stress visits. This points to the potential unreliability of using standard statistical methods and salivary cortisol alone as an objective measure of acute stress.” 

Further, cortisol is not always convenient to measure in free-living subjects due to the need to collect saliva samples, a mildly invasive protocol, and to account for diurnal variations. These challenges, especially as they relate to pregnancy, have been outlined in lines 117-120. Conversely, HRV is increasingly feasible to measure both in controlled and free-living settings with non-invasive wearable technology. We have added a sentence to the introduction explaining this point (line 128-130):

“HRV can be easily measured using non-invasive wearable technology, making it an attractive physiological measure in both lab-based and free-living research settings. However, little is known about the reliability of HRV parameters as a measure of stress and stress reactivity in pregnancy, or how HRV performs compared to cortisol levels to characterize stress.”

While HRV may be a preferred measure in free-living research studies over more invasive and complicated saliva collections for cortisol, our study demonstrates that the sensitivity for detecting stress using HRV in the absence of cortisol requires further work. HRV is a continuous measure and helps to complement cortisol which has temporal resolution. In this way, HRV helps to fill the gaps between cortisol measurements and as machine learning research develops in this field, HRV could be used to help interpolate cortisol responses as an indicator of stress. This point is explained in the original version of the discussion (lines 471-477):

“Further, cortisol cannot be easily measured continuously but only at specified time points of sample collections. Thus, it would benefit future research in the field of stress to computationally predict an individual’s pattern of cortisol response to stress using other physiological bio signals measured continuously by wearable sensors, such as HR, HRV parameters, GSR, and respiration rate. This could help reduce the need to measure salivary cortisol in future research, thereby increasing efficiency and reducing the burden for participants and researchers.”

Comment #13: Unclear what is meant by ‘discrimination stress’ line 403. Was this in the study referred to? 

The referenced study measured maternal stress related to experiences of racial discrimination. We have updated the sentence for clarity as follows (line 440-442):

“They found that higher self-reported levels of discrimination stress related to experiences of discrimination among participants were associated with lower levels of cortisol reactivity and higher levels of RMSSD following the TSST”.

Comment #14: Line 445 – was baseline stress measured in the chosen population? If not I’m not sure this argument is very strong. 

Response: Baseline perceived stress levels were assessed in this cohort at the beginning of the study. A statement of this measure has been added to the methods under Data Collection (line 173-175):

“On visit 1, usual stress levels over the past month were assessed by the Perceived Stress Scale.”

We have reported the average stress scores in the beginning of the results section (line 294-296) 

“The mean PSS score was 13.4±3.7 out of a potential range of 0-40. PSS values from 14-26 indicate moderate stress and values from 27-40 indicate high stress levels. Thus, on average, this population of pregnant women had borderline moderate levels of perceived stress in their daily lives.”

We have also expanded on the relevance of this measure in the discussion in the Strengths and Limitations section (line 485-487):

“While our cohort perceived borderline moderate levels of stress over the past month, it is unknown whether such baseline stressors contribute to higher or lower stress responses to the controlled stress task compared to non-Hispanic women.”

REVIEWER #2: 

Comment #1: Technical soundness:

The medical results of this experiment are limited to pregnant women of Hispanic origin who were overweight before pregnancy. There seem to be minor flaws in the collection of data, as described below. The attempt to build an objective stress assessment model is interesting and commendable, with an acceptable accuracy (average 77%), although visual assessment of graphs classified 86% of cases correctly. Medically, it is more interesting to objectively detect and evaluate chronic stress exposure rather than acute stress.

Response: We agree with the reviewer regarding the importance of chronic stress evaluation for translational science purposes. However, this paper provides groundwork for future studies to consider whether HRV in free-living settings alone is adequate to detect the presence of stress or if the combination of cortisol and HRV measures may be a preferred combination to increase accuracy. Whether this combination is also applicable to chronic stress assessment remains to be determined, and we have added a sentence at the end of the conclusion paragraph to highlight this gap for future studies in the field (line 530-531):

“The potential translation of this work to the assessment of chronic stress in humans requires further research.”

Comment #2: This is a secondary analysis of a cross-over study which primarily intended to study the effect of psychological stress on postprandial metabolic response to a standardized meal. The metabolic data are not accounted for in the paper. Supposedly they are subject for a different publication.

Strictly speaking, therefore, the intervention in this experiment is primarily the ingestion of a standardized meal (which normally triggers the parasympathetic autonomic system) and, secondly, a stress test task (which activates the sympathetic autonomic system). The superimposed triggers of the autonomic nervous system of the experiment complicate the interpretation of data and might explain some of the heterogeneity noted. The authors might elaborate on the effect of the dual intervention in their text.

Response: We thank the reviewer for this valuable comment. The parent study required meal ingestion to address the primary research question regarding the metabolic effects of the interplay of nutrition and stress. We acknowledge that the meal ingestion may limit interpretation of results in the present paper as it is unknown from our data how the physiological stress measurements may have differed if participants remained in the fasting state for the duration of each visit. However, given that all participants consumed the same meal at approximately the same time on each visit, we would assume that any potential stimulation of the parasympathetic response is similar across subjects. Thus, the difference in the physiological stress response between visit 1 and visit 2 is standardized in this cohort, allowing the machine learning algorithm to systematically detect this difference in physiological measures. We have added a discussion around this point in the Strengths and Limitations section (line 498-503):

“Introduction of meal ingestion as part of the parent study could also be considered a limitation for the present analysis, as it is unknown to what degree the act of consuming a meal may have moderated the stress response. However, as the meal type and time of ingestion was standardized across all subjects and occurred on both visits, we assume that any potential influence of the meal on the physiological stress response is also standardized across participants.”

Comment #3: The experiments in the study seem to have been conducted rigorously, except for the fact that visit 2 (TSST visit), other than introducing the intended psychological stress, inadvertently added physical movements (the women had to walk to a separate room) which might have increased heart rate and therefore influenced also the HRV parameters. The authors comment appropriately on this in the discussion section (line 368).

Response: We agree that this added movement during visit 2 could have influenced the HR and HRV features and have acknowledged this in the discussion, as noted by the reviewer.

Comment #4: Another concern, which is not commented on, is that the time of awakening of the participants are missing and might have been different on the two occasions. Time of awakening is crucial to the assessment of cortisol levels, since the time from the morning peak values of cortisol would have influenced the baseline cortisol values, and might not be comparable between the two occasions (although the participants arrived at the research centre at about the same time). If most of them rose later in the morning on visit 2, for example, they would indeed display higher cortisol levels from start. It would therefore probably have been more correct to analyse changes from baseline of cortisol rather than absolute cortisol levels and AUC. The level of stress in the morning in the participants’ everyday home environment, before arriving at the research centre, are not taken into account and might also be of importance for baseline cortisol levels

Response: We agree with the reviewer that time of awakening and earlier morning stressors could have influenced cortisol levels during the study visits and we regret this oversight in the original submission. We did ask participants to report their time of awakening on the morning of each study visit. In this revision, we have computed the time interval in minutes from time of awakening until time of arrival at each visit and regressed each participant’s cortisol values against this time interval variable at each visit to generate standardized cortisol values (i.e., adjusted for time of awakening). We then repeated the machine learning models and statistical analysis and updated the results accordingly in the manuscript. We note that the paired t-test for difference in cortisol AUC between visits is no longer significant using the standardized cortisol values (Table 2), however, the outcome of the machine learning model remains virtually unchanged (Table 3). 

We have added a sentence in the methods to state that time of participant awakening was recorded (line 173-174):

“Participants were asked to report their time of awakening on the morning of each visit.”

In the methods section, we have added a description for how the cortisol values were adjusted (line 213-216):

“To account for inter-individual variation in cortisol values influenced by time of awakening on the mornings of the study visits, we used normalized cortisol variables, adjusted for time (minutes) interval from awakening until time of arrival at the visit, as the input for statistical analysis and the assessment model.”

Unfortunately, we are unable to adjust the cortisol values for any potential stressful events that occurred before arriving to the visits, and have acknowledged this as a limitation in the discussion (line 503-307):

“While we adjusted cortisol values for time of awakening at each visit, it is also possible that stressful experiences occurred for some participants before arriving to their study visits, which could have elevated their baseline cortisol values. We were unable to account for these potential sources of external stress in the data.”

Comment #5: A technical weakness, which the authors already point out in the text (line 454), is that the Actiheart registration did not provide a raw ECG-signal, which risks to make the HRV data less accurate.

Line 202: Based on what criteria were Actiheart artifacts removed (since no visual inspection of ECG was possible)? 

Response: We wish to clarify that the abnormal IBI and HRV values generated by motion artifacts were removed before proceeding with the analysis. We did not have access to the raw ECG signal. The Actiheart only removed part of the motion artifacts but we found that some IBI data was still affected by noise. Therefore, we removed the abnormal IBI and HRV variables according to the IBI removal criteria introduced in a previous HRV validation study. The methodology has been updated and the corresponding study cited (line 224-227):

“The abnormal IBI and HRV values generated by motion artifacts were removed before proceeding with the analysis according to the removal criteria described in another study utilizing HRV measures []. The removal criteria are based on the normal range of HR and IBI values.”

Comment #6: Statistics

The time t used for calculation of the AUC seems to be the same for cortisol and for HRV-parameters (namely 120 minutes). This probably suits cortisol dynamics which are quite slow, but does not match HRV dynamics, since parasympathetic withdrawal and reinstatement are almost immediate in effect, and the researchers will therefore in the same lapse of time capture instances of stress (with lower HRV), but possibly also a period of relaxation/ relief after stress, which might translate into higher HRV, thus leveling out the AUC on visit 2 (particularly of the parasympathetic indices). This might explain why the researchers found no statistical difference in HRV parameters when comparing AUCs. It probably does not affect the objective stress assessment model, however.

Response: Thank you for this important insight. We acknowledge that the ideal approach would be to compute HRV reactivity scores (i.e. change from baseline to peak stress moment at ~45 mins) for the various HRV parameters, and compare these reactivity scores between visits in the paired t-tests. Unfortunately there was very limited baseline HRV data collection performed before starting the stress/non-stress task in this study (often less than 5 minutes) and for some subjects, several of these baseline data points had to be removed due to noise. While we acknowledge this as a limitation, we agree with the reviewer that the machine learning approach overcomes this challenge by using all data available from the ECG rather than relying on suboptimal summary measures. We have added this limitation to the discussion section (line 511-515):

“Most of these gaps occurred at the very beginning of data collection, in the 5 minutes before participants underwent the stress or non-stress task period. This resulted in inadequate baseline HRV measurements from which to compute reactivity scores for each parameter, which may have been a more robust summary measure than the AUC values on which we relied for the statistical analysis.”

Comment #7: This reviewer has no deeper insight in programming or in machine learning-based algorithms and can therefore only comment on the quality of medical data used for feature selection. To this reviewer’s knowledge all HRV parameters are influenced by HR, are highly interdependent and are not normally distributed. HR and AVNN are inherently redundant - it is surprising the feature selection allows both, considering it is constructed to avoid overfitting?

Response: We thank the reviewer for this comment. During the feature selection process, the algorithm tries all different combinations of features and the contribution to the decrease in the impurity of each feature is computed. The importance of each feature is ranked according to this contribution. As we can see from the feature selection results, a combination of 5 features including AVNN, HR, SDNN, LF, and LF/HF are selected when cortisol is not added. AVNN and HR are the first two features which means they each make a bigger contribution to discriminate stress vs. non-stress than the remaining three features. HR represents the frequency of heart beats and AVNN represents the average length of heartbeat intervals. Although AVNN and HR seem highly related from a statistical perspective, they represent different HRV features and thus, both add value to the model. However, after cortisol is added, HR is no longer selected as an important feature but AVNN remains. We have added an explanation about these features to the results section for further clarity (line 346-349):

“HR represents the frequency of heart beats and AVNN represents the average length of heartbeat intervals. Although AVNN and HR appear highly related from a statistical perspective, they represent distinct HRV features and thus, both contribute value to the model”.

Comment #8: All data supporting the conclusions seem to be fully available in the Supporting information file. (This is a secondary analysis of a cross-over study which primarily intended to study the effect of psychological stress on postprandial metabolic response to a standardized meal. The metabolic data are not accounted for in the paper. Supposedly they are subject for a different publication.)

Response: In this resubmission, we have uploaded our final dataset of ECG and cortisol data from all participants at each visit to the online data repository, DRYAD. The dataset in DRYAD will be publicly available upon acceptance of the manuscript.

This work is presented in intelligible standard English.

Comment #9: Line 108: Statement should be backed up by a reference.

Response: A reference has been added to support this sentence (line 120).

Comment #10: Line 172 and 178: TSST reference?

Response: References have been added to support these statements (lines 194 and 198).

Comment #11: Line 216: “,” instead of “.” as in one sentence.

Response: This has been updated (line 242).

Comment #12: Line 453: Actiheart instead of Actiheat.

Response: This has been updated (line 502).

Comment #13: Figure 5: Dots should not be connected by lines, and it should be clarified that unlike Figure 2, 3 and 4, x is no longer time, but represent discrete individuals.

Response: We have updated Figure 5 and now present the data in bar chart format for improved interpretability. Each x-axis is labeled such that each set of bars represents unique participants.

---

## [Decision Letter · Decision Letter 1]

26 Aug 2022

Prenatal Stress Assessment using Heart Rate Variability and Salivary Cortisol: A Machine Learning-Based Approach

PONE-D-22-11525R1

Dear Dr. Lindsay,

We’re pleased to inform you that your manuscript has been judged scientifically suitable for publication and will be formally accepted for publication once it meets all outstanding technical requirements.

Kind regards,

Enzo Pasquale Scilingo, Ph.D.

Academic Editor

PLOS ONE

Additional Editor Comments (optional):

Reviewers' comments:

Reviewer's Responses to Questions

**Comments to the Author**

1. If the authors have adequately addressed your comments raised in a previous round of review and you feel that this manuscript is now acceptable for publication, you may indicate that here to bypass the “Comments to the Author” section, enter your conflict of interest statement in the “Confidential to Editor” section, and submit your "Accept" recommendation.

Reviewer #1: All comments have been addressed

2. Is the manuscript technically sound, and do the data support the conclusions?

Reviewer #1: Yes

3. Has the statistical analysis been performed appropriately and rigorously? 

Reviewer #1: Yes

4. Have the authors made all data underlying the findings in their manuscript fully available?

Reviewer #1: Yes

5. Is the manuscript presented in an intelligible fashion and written in standard English?

Reviewer #1: Yes

6. Review Comments to the Author

Reviewer #1: (No Response)

7. PLOS authors have the option to publish the peer review history of their article (what does this mean?). If published, this will include your full peer review and any attached files.

Reviewer #1: No

---

## [Editor Report · Acceptance letter]

1 Sep 2022

PONE-D-22-11525R1 

Prenatal Stress Assessment using Heart Rate Variability and Salivary Cortisol:  A Machine Learning-Based Approach 

Dear Dr. Lindsay:

I'm pleased to inform you that your manuscript has been deemed suitable for publication in PLOS ONE. Congratulations! Your manuscript is now with our production department. 

Kind regards, 

on behalf of

Professor Enzo Pasquale Scilingo 

Academic Editor

PLOS ONE